# The influence of time pressure on translation trainees' performance: Testing the relationship between self-esteem, salivary cortisol and subjective stress response

**Ana Mª Rojo López**[1]*, **Paula Cifuentes Férez**[1]*, **Laura Espín López**[2]*

1 Department of Translation and Interpreting, Faculty of Arts, University of Murcia, Murcia, Spain,
2 Department of Human Anatomy and Psychobiology, Faculty of Psychology, University of Murcia, Murcia, Spain

* anarojo@um.es (AMRL); paulacf@um.es (PCF); lespin@um.es (LEL)

**Data Availability Statement:** The data underlying this study are available on the OSF repository (https://osf.io/ku6cz/).

## Abstract

Translators face hectic daily schedules with deadlines they must duly meet. As trainees they receive tuition on how to work swiftly to meet them efficiently. But despite the prominent role of time pressure, its effects on the translation process are still scarcely researched. Studies point to the higher occurrence of errors under stringent time constraints. Most of these studies use key-logging or eye-tracking techniques to identify the problems encountered. But no attempt has yet been made to measure the physiological effects of time pressure in English-to-Spanish translation and their interplay with trainees' psychological state. The present study researches the influence of time pressure on translation by exploring trainees' physiological response (i.e., salivary cortisol) and psychological traits (i.e., self-esteem and anxiety). 33 Spanish translation trainees translated 3 English literary texts under different time pressure conditions: Text 1 (no time limit), Text 2 (10 minutes), Text 3 (5 minutes). Regression analysis results showed that higher cortisol levels during preparation predicted higher number of meaning errors in Text 1 and lower number of translated words in Text 2 and 3. Besides, higher trait anxiety emerged as predictor of lower number of translated words, but higher accuracy under extreme time constraints and in the absence of time pressure. Higher self-esteem correlated with lower levels of anxiety and lower levels of cortisol during preparation and recovery, suggesting that it may act as a protective factor against stress. And yet, the regression analysis showed that higher self-esteem predicted lower meaning and total accuracy under extreme time pressure. Besides, in our correlation analysis self-esteem was positively related to the number of translated words in Text 2 and 3. Results suggest that even if self-esteem could be a protective factor against stress, it may also have a negative effect on task performance mediated by overconfidence.

**Funding:** This work was supported by the Spanish Ministerio de Ciencia, Innovación y Universidades, Agencia Estatal de Investigación and FEDER/UE funds (grant number FFI2017-84187-P). The funders had no role in study design, data collection and analysis, decision to publish, or preparation of the manuscript.

**Competing interests:** The authors have declared that no competing interests exist.

# Introduction

Time equals money for most professionals, but even more so for translators, who frequently work under stringent time constraints. Wasting time is rarely an option for translators, who most commonly work with deadlines that are far too short and contract prices that are far too low [1]. The pressure on cost and productivity imposed first by the 2017 global recession and later by the most recent COVID-19 pandemic has aggravated the situation, contributing to the proliferation of freelance work and the technologization of translation labor [2]. The time-money equivalence is so prominent in translation that working under time pressure and coping with its psychological consequences has become one of the golden rules of the translator's job.

A central question, therefore, is to establish the psychological consequences of time pressure and its effects on translation performance. Tight deadlines and telework may not be necessarily negative for productivity or psychologically harmful, but when the pressure perceived is too high or the exposure too long, they are likely to result in stress, affecting workers' performance as well as their psychological and physical wellbeing. The notion of perceived time pressure refers to the feeling that there is not enough time available to perform or complete a task [3]. In modern society, this perception has become a world standard to the point that acute time pressure has been identified as a physical and psychological stressor with a number of negative effects, such as tension, fatigue and general lower health and life satisfaction [4]. But there is also evidence pointing to positive consequences of time pressure for social behavior, altruism or honesty [5]. Time pressure may have positive or negative outcomes, depending on factors such as duration, context or individual perceptions, particularly whether it is regarded as motivating or as something to endure [6]. The mixed effects of time pressure are also related to non-conclusive evidence on the influence of stress on performance. Even if the effects of stress on performance are generally detrimental, there is evidence suggesting that when the level of stress is not too high, stress (also positively referred to as Eustress) can improve concentration and motivation [7]. Besides, results from neurophysiology point to the positive effect of brief acute stress for keeping the mammalian brain alert [8].

As for the effects of time pressure on task proficiency, work on decision-making shows that the presence of a time constraint in any math or problem-solving situation can influence performance, although the effects can also be mixed: the presence of time constraints could either foster engagement with the task or increase choice of the wrong strategy [9, 10]. However, the causal mechanism of this suboptimal strategy selection is still to be determined, since it is unclear whether time pressure interferes with strategy selection or simply impedes the optimal strategy due to an overload of working memory resources [11]. Specifically, pressure is expected to lead to worry, concern and other distracting thoughts about performance, which consume working memory resources [12]. Research also points to the concurrent effect of trait anxiety on overloading working memory and impairing performance [13]. In contrast, having no time restrictions to complete a task enables participants to focus attention on important task features, allowing them to choose the best strategy [14].

Despite the prominent role of time pressure on decision-making and performance, its effects on the translation process are still barely researched. Existing studies provide evidence for the higher occurrence of errors under stringent time conditions by focusing on the effects of time pressure on different stages of the translation process [15, 16] or product quality [17, 18]. Many of these studies use key-logging [e.g., 15–17, 19] or eye-tracking techniques [e.g., 20] to identify and describe the problems encountered in the different time conditions. But no attempt has yet been made to measure the physiological effects of time pressure and their interplay with translators' psychological state and trait profiles. On the whole, the extant

literature provides evidence for the higher occurrence of errors under stringent time conditions, but results are not conclusive. Time pressure does not always result in more errors [17, 21] and enough time does not always guarantee less errors [22, 23].

Another key question, thus, is to establish the factors that mediate the relationship between time pressure and translation quality. Amount of available time and number of words to be translated certainly matter. Evidence shows that time influences not only translation quality, but also translation behaviour. Results reveal that translation quality decreases more than 15% when translators are asked to translate more than 200 words in 10 minutes [17]. In contrast, when too much time is available, translators increase the number of problem-solving activities [15, 16]. Time also influences translators' allocation of attention to either the source text or the target one, with evidence showing that reading-for-comprehension of the source text is easier to adapt to variable time constraints than reading-and-monitoring of the target text [20].

Besides available time, data suggest that other contextual factors, such as task difficulty, and personality traits may also play a role in the perception of time pressure. The influence of task difficulty has been widely documented in some interpreting modes, where the message must be rendered orally in the target language with tight time constraints [e.g., 24]. Some features of certain modes of remote interpreting, such as the use of booths in conference interpreting or the lack of presence in telephone interpreting may additionally increase perceived task difficulty and time pressure [25–27]. A number of personality traits (e.g., especially those related to self-confidence, such as self-esteem, trait anxiety or perfectionism) are also likely to be of importance in modulating perceived time shortage. Evidence from psychology and health studies points to an effect of time pressure and task difficulty on confidence: time pressure increases confidence in easy cases but reduces confidence in difficult cases [28]. It seems that time pressure increases the negative effect of overconfidence, which usually leads people to overestimate their actual performance in difficult tasks, but underestimate it in easy ones [29]. Related evidence from studies on competitive sporting situations suggest that while high levels of competitive trait anxiety (CTA) can result in impaired performance, lower levels can be beneficial [30]. Moreover, lower levels of CTA have been related to other traits potentially beneficial for performance, such as higher confidence, perfectionism or lower levels of state anxiety [31–33].

To our knowledge, no study has specifically researched the influence of personality traits in modulating perception of time pressure in translation and interpreting, although previous results on the influence of time pressure on translation quality pointed to the role of individual differences. In De Rooze's (2003) study, for instance, 25% of the participants still manage to produce high quality translations even under acute time pressure conditions. Besides, some studies have explored the role of traits such as emotional stability, intuition, resilience, self-esteem or openness to experience in predicting quality performance in sign language interpreting and translation [34–40].

The study introduced in the next section explores the role of self-esteem in translation under different time pressure conditions. Previous results on self-esteem in translation performance suggest that high self-esteem levels may lead to more mistakes, at least on spelling and punctuation aspects [38]. In psychology, self-esteem has been related to mental health outcomes, particularly to subjective stress reaction. Stress is triggered by situational demands that exceed an individual's cognitive resources [41], so their attitude toward coping with such situations may influence their stress reaction. Self-esteem refers to a person's subjective judgement of their own worth or adequacy, and self-acceptance [42]. It is a valuable psychological resource, since high levels of self-esteem are a resilience factor, protective against adverse mental health outcomes [43, 44] and stressful events–for instance, high levels of psychological resources (including self-esteem) have been related to lower reactivity to stress [44, 45].

Nevertheless, results on the effect of self-esteem on cortisol response and task performance are not conclusive. Low levels of self-esteem are mostly related to inferior performance and heightened cortisol reactions to achievement stress [46–48], but there is also evidence on a positive correlation between self-esteem and cortisol response on university students mediated by the need for social approval [49]. Regarding task performance, studies have failed to demonstrate that high self-esteem leads to good task performance, with the exception that it facilitates persistence after failure [50].

Previous translation studies on the influence of time pressure have emphasized the effect of time constraints on professionals' and students' performance, assuming the intervening role of stress. However, to date no attempt has been made to measure the physiological and psychological effects of time constraints in English-to-Spanish translation. The present study addresses this gap by exploring the interplay between translation students' self-reported levels of self-esteem and trait anxiety and their objective (i.e., salivary cortisol response) and subjective (i.e., self-reported state anxiety) response to the stressful situation of translating under tight pressure conditions.

Based on the previously discussed research, the following hypotheses are posed: (1) when translating under time pressure students' cortisol levels will be higher and their performance will be worse than under no time pressure; (2) under time pressure constraints (i.e., text 2 and text 3), lower levels of self-esteem will be associated with heightened cortisol reactions, higher levels of state anxiety and more mistakes. Conversely, higher levels of self-esteem will be associated with attenuated reactivity to acute time pressure, lower levels of state anxiety and less mistakes, except for spelling and punctuation errors; (3) under time pressure constraints (i.e., text 2 and text 3), higher levels of self-reported anxiety will be associated with heightened cortisol reactions and poorer performance.

## Materials and method

The study had a within-subject design. Each participant translated three comparable literary texts from English into Spanish in the same order across three different time-constrained conditions: from the no time pressure condition of Text 1, through the low or medium time-pressure condition of Text 2 (10 minutes for 153 words), to the high time-pressure condition of Text 3 (5 minutes for 153 words). Time-pressure conditions were not randomized to ensure the progressive build-up of time pressure and avoid time limitations on cortisol response (see Procedure below). Presentation of texts was not randomized either to rule out extraneous variables related to the specific translation difficulties of each text.

### Participants

An initial sample of 70 undergraduate students from the Translation and Interpreting Degree at the University of Murcia (Spain) volunteered to take part in the study. The inclusion criteria were established by completion of an online questionnaire requesting general and health information on psychological disorders, cardiovascular diseases, endocrine disorders, asthma or smoking habits (more than 10 cigarettes per day). Of the initial sample, 25 were excluded due to any of these criteria, and 12 subjects were finally eliminated due to problems during the experimental procedure. Therefore, the final sample was composed of 33 translation students, who were single, had no known medical or psychological problems and ranged from 19 to 20 years of age. Their mean age was 19.36 years (S.D. = 0.57). The sample had a majority of female participants (87%, n = 29), which was representative of the share of women found in the total population of the degree. Females were asked for their phase of the menstrual cycle, but most answers were tentative, so this data was not included in the study. Of the total of 29 women,

**Table 1. Descriptive characteristics of the sample.**

| Variables | Minimum | Maximum | Mean | S.D |
|---|---|---|---|---|
| Age | 19 | 20 | 19.36 | 0.48 |
| BMI[a] | 16.65 | 26.51 | 21.29 | 2.60 |
| STAI Trait[b] | 8 | 53 | 28.97 | 10.58 |
| RSES[c] | 19 | 40 | 28.64 | 5.99 |

[a]Body Mass Index.

[b]Trait anxiety.

[c]Rosenberg Self Esteem scale.

only 8 were taking the contraceptive pill. They were all second-year students at the time of the experiment. For all of them Spanish was their mother tongue and English their main foreign language. The main characteristics of the sample are shown in Table 1.

All participants granted their consent according to the Declaration of Helsinki, and the protocols were approved by the University of Murcia Ethics Committee. Participants were informed of the general purpose of the study and were told that they could leave the experiment at any point. They were awarded with extra 0.5 points in one of their spring semester modules, and were rewarded with snacks after completion of the experiment.

## The source texts and translation performance measures

Three English literary texts were selected from the novel *The Ballroom* (2016) by Ann Hope (see S1 File). They were all of similar difficulty according to the Flesch-Kincaid the Reading Ease and Grade Level, the Gumming Fog Score, the SMOG Index, and the Coleman Liau Index (see Table 2).

The three texts were descriptive and had no dialogues. They were also comparable in length: the first text (Text 1) contained 150 words, whereas the second (Text 2) and the third (Text 3) had 153 words each. Texts shorter than 200 words were selected for two main reasons: to avoid participants' fatigue, since they were asked to translate the three texts to allow for within-subject comparisons, and to differentiate between a condition (Text 2) where the task was affordable within the given time limit (10 min.) (see for instance [17]), and one (Text 3) where the time limit (5 min.) imposed very tight time constraints.

As for translation performance measures, the accuracy of translated texts was assessed in terms of number of errors (see Table 3). From a total score of 10 points, we subtracted from 0.25 to 1 point for each error, depending on their type. The evaluation sheet was adapted from the one designed for the TRANSCREA research project to assess both the accuracy and creativity of translation [51]. Accuracy was thus assessed by computing errors on three different categories: transfer of meaning (including false meaning, opposite meaning and unnecessary

**Table 2. Scores of text difficulty, grade conversion and comprehension of the three texts.**

| | Text 1 | Text 2 | Text 3 | Grade Conversion—Comprehension |
|---|---|---|---|---|
| **Flesch Kincaid Reading Ease** | 92.7 | 100.4 | 96.4 | 5th Grade—Very easy to read |
| **Flesch Kincaid Grade Level** | 3 | 2.5 | 2.3 | 5th Grade—Very easy to read |
| **Gunning Fog Score** | 5.6 | 4.1 | 4.9 | 5th Grade and below—Very easy to read |
| **SMOG Index** | 3.9 | 2.7 | 3.2 | 5th Grade—Very easy to read |
| **Coleman Liau Index** | 9.6 | 7.3 | 7.6 | 8th, 9th & 10th Grade—Conversational English |

**Table 3. Evaluation sheet for the assessment of target texts.**

| Transfer of meaning | |
|---|---|
| False meaning / Not the same meaning | – 0.5 |
| Opposite meaning / Incoherent meaning | – 1 |
| Unnecessary omission / addition of meaning | – 0.5 |
| **Transfer of pragmatic function** | |
| Loss of cultural reference and/or implied meaning | – 1 |
| Loss of humor or irony | – 1 |
| **Correctness** | |
| Grammatical errors | – 1 |
| Cohesion errors (connectors, loss of repetition) | – 0.5 |
| Orthotypographic errors: | |
| Typos | – 0.25 |
| Written accents and punctuation marks | – 0.5 |
| Serious spelling mistakes | – 1 |

omissions or additions); transfer of pragmatic function (including mainly loss of literary style, cultural references, implied meaning, humor or irony); and correctness (including grammatical errors, errors in the cohesion of the text, typos and punctuation and spelling errors).

The number of translated words was also computed across the different translation tasks by counting the number of words translated by each participant in each source text. All participants completed the translation of Text 1, but many did not finish Text 2 and 3, due to the time restrictions imposed. A correction index was thus applied to the scores for these two texts in order to increase comparability among the error categories across the three different texts. The correction index consisted of dividing the score for each category of error by the number of translated words and multiplying the result by the total number of words from the source text (i.e., 153 words for both Text 2 and Text 3).

## The self-report measures

Self-esteem was measured by the Spanish version of the Rosenberg Self-Esteem Scale (RSE) [52]. The Rosenberg Self Esteem Scale (RSES) [53] is commonly used and its internal consistency and reliability were confirmed in many previous studies [54]. It comprises 10 statements. Participants rate the extent to which they agree with each statement on a four-point Likert scale, (0) strongly disagree to (4) strongly agree for items 1, 2, 4, 6 and 7 and opposite rating for items 3, 5, 8, 9 and 10. A total score is obtained by adding up all responses and may range from 0 to 30, with higher scores indicating higher self-esteem [55]. The Spanish scale has a Cronbach's alpha ranging from 0.85 to 0.88, and the alpha value for the scale in our sample was 0.76.

Anxiety was measured by the Spanish version of the State-Trait Anxiety Inventory (STAI) [56, 57]. This is a 40-item self-report inventory that measured participants' levels of state anxiety (STAI-S) and trait anxiety (STAI-T) on a 4-point Likert scale, ranging from 1 (almost never) to 4 (almost always). The adapted scale has a Cronbach's alpha ranging from 0.90 to 0.93 [57]. In our sample, alpha values were 0.92 for trait anxiety and 0.94 for state anxiety.

## The physiological biomarker

Salivary cortisol was collected using the Salivette® sampling device (Sarstedt, Newton, NC). Five saliva samples were collected over a 60-minute period at five different points in time with

reference to the start of the experimental task (sample t0): t-20 (baseline, 20 mins. before the start of the experimental task), t0 (the start of the experimental task), t+20 (20 mins. after the start of the task), t + 35 and t+45.

Participants were instructed to place the cotton swab inside their mouths for 2 minutes, not to chew the cotton because it may affect salivary protein composition as well as the flow rate [58], and move the swab around in a circular pattern to collect saliva from all the salivary glands [59]. The uncentrifuged saliva samples were stored at −80 ˚C immediately upon collection until analyses were performed. To reduce sources of variability, all five samples taken from each participant were analyzed in the same assay. The samples were analyzed by a competitive solid phase radioimmunoassay (tube coated), using the commercial kit Coat-A-Count Cort (DPC, Siemens Medical Solutions Diagnostics). Assay sensitivity was 0.5 ng/ml. Cortisol levels were expressed in nmol/l, with coefficients of intra- and inter-assay variations of less than 10%.

## Procedure

Participants meeting the criteria were contacted by telephone and asked to attend the experimental session, which took place in a quiet room at the Faculty of Arts at the University of Murcia. Before each session, participants were asked to maintain their general habits, sleep as long as usual, refrain from heavy physical activity the day before the session, and not to consume alcohol after the previous dinner. Instead, they were instructed to drink only water and not to eat or take any stimulants in the 2h prior to the session, including coffee, cola, caffeine, tea or chocolate. Care was taken not to evaluate participants during stressful periods (such as exam periods).

Experimental sessions were always held between 2pm and 6pm (the data gathering protocol is schematically presented in Fig 1). Participants were tested individually and in a single session. After arrival at the room, the experimenter asked the participants whether they had followed the instructions previously given. They were informed that the experiment involved completing some psychological tests and translating three texts under different time constraints. They were also told that five salivary samples would be collected at different times throughout the experimental session and explained how to use the salivette for the collection of salivary cortisol. Experimental sessions consisted of several phases of different duration. Each session took approximately 1h and 10 min to complete.

The 1st sample was then taken and they were asked to complete the two tests: the RSE and the STAI (both the STAI-T and the STAI-S); when completed, the 2nd salivary cortisol sample was taken just before the start of the translation task. No time limit was given for Text 1, but participants spent a maximum of 20 minutes in this translation. On completion of this translation the 3rd cortisol sample was taken.

They were then given 10 minutes to translate Text 2. No cortisol sample was taken after translating Text 2 because the time differences with the end of Text 1 and the end of Text 3 were not long enough to provoke significant cortisol responses (i.e., 10 minutes between the end of Text 1 and Text 2 and 5 minutes between the end of Text 2 and the end of Text 3). Considering that the maximum cortisol increase is observed 30 minutes post-stress [60, 61], we established 15 minutes as the minimum threshold to guarantee the increase.

When they finished translating Text 2, they were given 5 minutes to translate Text 3. The 4th salivary sample was collected on completion of the last translation. Lastly, participants were asked to complete again the STAI-state questionnaire and the 5th salivary sample was finally taken. A visible countdown timer was displayed in the room, but participants were also told to

## Experimental task

**Fig 1. Different phases of the experimental protocol.**

display one on their own computer screen (https://www.online-stopwatch.com/countdown-clock/full-screen/) to keep track of time. Participants were allowed to use their own computers and any online documentation resource they wished during the translation task in order to maximize ecological validity.

### Data analysis

Salivary cortisol was tested for normal distribution and homogeneity of variance using the Kolmogorov–Smirnov test before the statistical procedures were applied. These analyses did not reveal significant deviations from normality.

To assess sample differences in STAI-S before and after the translation tasks, as well as the cortisol response across the different phases of the protocol, we conducted separate repeated measures analyses of variance (ANOVAs) with time as a within-subjects factor (five phases for salivary cortisol: t-20, t0, t+20, t+35 and t+45 and two phases for anxiety: pre- and post-task).

Pearson's bivariate correlation analyses were conducted to test the relationship between self-esteem, trait-state anxiety and the five cortisol samples. They were also run to examine whether levels of anxiety, self-esteem and stress-induced cortisol were related to performance scores in the translation tasks. These correlations were tested for each type of error scale (the accuracy scale, i.e., the mean score obtained from the three subscales: meaning, pragmatic and correctness errors; and each of the three subscales separately) and for each text (Text 1, Text 2, Text 3). The number of words translated in each text was also included as a performance score.

Additionally, linear regression analyses were performed to test the relationship between translation performance in the three texts and baseline and preparation cortisol levels using t-20, t0 and state-trait anxiety and RSE as the predictor variables, and the translation error scales (mean value of score in each scale: accuracy, meaning errors and pragmatic errors) and number of translated words as the dependent variables, in each text.

# Results

## Salivary cortisol

A repeated-measures ANOVA was conducted with time (5) as within-subject factor to test differences in salivary cortisol between the different phases. The results did not show a significant main effect for this factor [$F_{(4; 128)} = 1.42$, $p = 0.22$, $\eta2p = 0.04$], revealing no statistically significant differences in participant's cortisol levels between the different phases (see Fig 2).

## State anxiety

Another repeated-measures ANOVA was conducted with time (2) as within-subject factor to test differences in STAI-S before and after the experimental task. The results showed a significant main effect for this factor [$F_{(1; 32)} = 3.92$, $p = 0.05$, $\eta2p = 0.10$], with higher scores of state anxiety in the post-task as compared with the pre-task.

## Self-esteem

Pearson's bivariate correlation analyses showed a significant negative correlation between self-esteem and STAI-T ($R = -0.80$; $p<0.001$); self-esteem and STAI-S pre ($R = -0.70$; $p<0.001$) and post task ($R = -0.76$; $p<0.001$); and self-esteem and t0 ($R = -0.41$; $p = 0.02$); and t+45 ($R = -0.41$; $p = 0.02$). Higher self-esteem was associated with lower STAI-T and STAI-S pre and post task and lower cortisol levels in t0 and t+45 (see Table 4).

To examine whether self-esteem, cortisol levels and anxiety could be associated with translation performance scores for the three texts, we conducted Pearson's bivariate correlation analyses between the variables. The results showed a negative correlation between t-20 and the

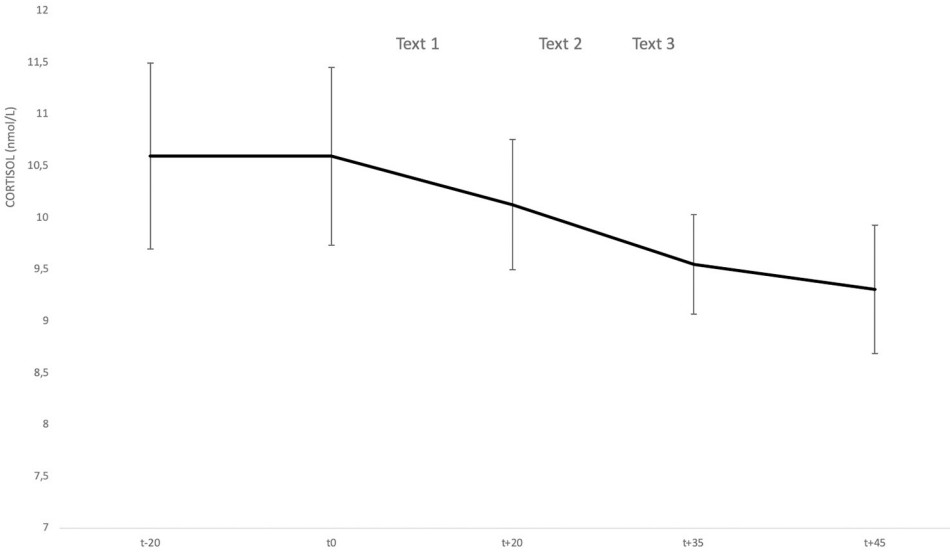

**Fig 2. Variation in cortisol response during the experimental session.**

**Table 4.  Pearson coefficients for associations between RSES and state-trait anxiety and cortisol levels.**

|  | 1 | 2 | 3 | 4 | 5 | 6 | 7 | 8 | 9 |
|---|---|---|---|---|---|---|---|---|---|
| 1. RSES[a] | - | | | | | | | | |
| 2. STAI-T[b] | **-.80**** | - | | | | | | | |
| 3. STAI-S-PRE[c] | **-.70**** | **.78**** | - | | | | | | |
| 4. STAI-S- POST[d] | **-.76**** | **.72**** | **.81**** | - | | | | | |
| 5. T-20[e] | -.32 | .27 | .28 | **.38*** | - | | | | |
| 6. T0[f] | **-.41*** | **.41*** | .32 | .29 | **.79**** | - | | | |
| 7. T+20[g] | -.19 | .27 | .19 | .19 | **.44**** | **.71**** | - | | |
| 8. T+35[h] | -.18 | .22 | .24 | .23 | **.44*** | **.60**** | **.72**** | - | |
| 9. T+45[i] | **-.41*** | **.40*** | **.46**** | **.41*** | .31 | **.49**** | **.42*** | **.65**** | - |

[a]Rosenberg Self Esteem Scale.

[b]Trait anxiety.

[c]State anxiety pre.

[d]State anxiety post.

[efghi]Cortisol samples.

* p < .05.

** p < .01.

number of translated words in Text 2 and Text 3. A negative correlation between t0 and the scores obtained on meaning and total accuracy for Text 1 and the number of translated words in Text 2 and Text 3. The cortisol level in t+20 correlated negatively with the score for meaning in Text 1, and the cortisol levels in t+35 and t+45 were positively correlated with the score for pragmatics in Text 3. STAI-T and pre-task STAI-S levels correlated negatively with the number of translated words in Text 2 and Text 3. Moreover, pre-task STAI-S levels correlated negatively with pragmatics in Text 1 and meaning in Text 2. Self-esteem levels correlated negatively with meaning and total accuracy in Text 3, and positively with the number of translated words in Text 2 and Text 3 (See Table 5).

Based on the results from the correlation analysis, a linear regression analysis was performed considering t-20, t0, pre-task state anxiety, trait anxiety and self-esteem (RSES) as the predictor variables, and text performance as the dependent variable.

In Text 1 higher cortisol levels in t0 were predictors of lower meaning (p = 0.01) and total accuracy (p = 0.01). Higher pre-task state anxiety was a predictor of lower pragmatic accuracy (p = 0.003), whereas higher trait anxiety predicted higher scores for the same category (p = 0.04).

In Text 2 higher cortisol levels in t0 were predictors of lower correctness (p = 0.05), and higher pre-task state anxiety of lower meaning (p = 0.005).

In Text 3 higher pre-task state anxiety was a predictor of lower correctness (p = 0.01) and total accuracy (p = 0.04). In contrast, higher trait anxiety was a predictor of higher correctness (p = 0.05), but less translated words (p = 0.04). Higher self-esteem predicted lower meaning (p = 0.05) and total accuracy (p = 0.02) (See Table 6).

## Discussion

This study investigated the influence of time pressure on translation performance by exploring the relationship between translation trainees' self-reported levels of self-esteem and anxiety and their hormonal and subjective responses to the stressful situation of translating under stringent time constraints as compared to translating without time restrictions.

**Table 5. Significant correlations between type of text and cortisol, anxiety and Rosenberg Self-esteem variables.**

| Type of text | | Cortisol | | | | | Anxiety | | | Self-esteem (RSES) |
|---|---|---|---|---|---|---|---|---|---|---|
| | | t-20 | t0 | t+20 | t+35 | t+45 | State Pre | State Post | Trait | |
| **TEXT 1** | Total Accuracy | -.18 | **-.44**** | -.31 | -.08 | -.17 | -.20 | -.08 | -.27 | .27 |
| | Meaning | -.15 | **-.41*** | **-.33*** | -.02 | -.02 | .01 | .008 | -.20 | .19 |
| | Pragmatics | -.13 | -.12 | -.05 | -.17 | -.32 | **-.44*** | -.24 | -.13 | .21 |
| | Correctness | .10 | -.09 | .04 | .12 | -.07 | -.14 | .08 | -.19 | .005 |
| | Translated words | - | - | - | - | - | - | - | - | - |
| **TEXT 2** | Total Accuracy | -.02 | -.17 | -.12 | .04 | -.07 | -.29 | -.13 | -.20 | .17 |
| | Meaning | -.12 | -.14 | -.08 | .02 | -.04 | **-.44**** | -.33 | -.17 | .20 |
| | Pragmatics | .16 | -.007 | .000 | .11 | -.08 | .17 | -.23 | -.10 | .01 |
| | Correctness | .04 | -.16 | -.25 | -.22 | .08 | .11 | .08 | .07 | -.08 |
| | Translated words | **-.36*** | **-.50**** | -.22 | .01 | -.16 | **-.36*** | .21 | **-.48**** | **.46**** |
| **TEXT 3** | Total Accuracy | .20 | .12 | .13 | .22 | .21 | .05 | .18 | .23 | **-.40*** |
| | Meaning | .30 | .17 | -.32 | -.09 | -.17 | .18 | .21 | .13 | **-.33*** |
| | Pragmatics | .10 | .10 | .09 | **.37*** | **.38*** | -.05 | .01 | .15 | -.22 |
| | Correctness | -.18 | -.14 | .10 | -.15 | -.20 | -.14 | .01 | .13 | -.09 |
| | Translated words | **-.46**** | **-.52**** | .01 | -.02 | .009 | **-.36*** | -.33 | **-.53**** | **.42*** |

* Significant at $p < .05$ level;

** Significant at $p = .01$ level.

Our first hypothesis predicted that when translating under time pressure students' cortisol levels will be higher and their performance will be worse than under no time pressure. However, results from the repeated-measures ANOVA did not confirm our hypothesis about the influence of time pressure on cortisol response since no statistically significant differences in participant's cortisol levels were reported between the different phases. Moreover, cortisol response displayed the opposite pattern to an average stress response, with tighter time constraints resulting in a progressive decrease in cortisol (See Fig 2). Cortisol response started to decrease right from the start of the translation through the whole task, even when translating the text with the tightest time constraints (i.e., Text 3, t+35). Since the effect of the stressor was confirmed by statistically significant pre/post- task differences in the trainees' levels of state anxiety, the decreasing pattern found in cortisol response suggests an attentional response to the task, which possibly increased as they engaged with the translation. This explanation aligns with previous evidence on the embodiment of narrative engagement which reports attentional focus to be related to lower levels of autonomic activity [62], and also with extant results on translation studies which point to a decrease in cortisol response and affective engagement caused by attentional focus and increased focal-task engagement [27, 63].

Regarding the relationship between cortisol response and performance scores, our correlation and regression analyses partially confirmed our hypothesis, since higher cortisol was associated with less accuracy and less translated words, but not in every condition. The correlation analysis showed that higher pre-task cortisol levels (t0) were associated with less meaning and total accuracy during the condition without time constraints (Text 1). These data were confirmed in the subsequent regression analysis, which revealed that higher pre-task levels of cortisol (t0) were predictors of lower meaning and total accuracy in this condition (Text 1), and also of lower correctness in Text 2. Moreover, the correlation analyses also revealed that pre-task levels of cortisol (t-20 and t0) were negatively associated with number of translated words in the conditions with time restrictions (i.e., Text 2 and Text 3), and that those trainees who

**Table 6. Linear regression analyses, considering t-20, t0, state- anxiety pre, trait-anxiety and RSES (Rosenberg Self Esteem scale) as the predictor variables, and text performance as the dependent variable (* = p ≤.05).** Beta, typified coefficient.

| Performance | Predictors | Text 1 | | Text 2 | | Text 3 | |
|---|---|---|---|---|---|---|---|
| | | Standardized β | p-value | Standardized β | p-value | Standardized β | p-value |
| **Total Accuracy** | t-20 | 0.48 | 0.09 | 0.37 | 0.22 | 0.36 | 0.18 |
| | T0 | -0.78 | **0.01*** | -0.42 | 0.19 | -0.33 | 0.24 |
| | State Anxiety pre | -0.06 | 0.82 | -0.44 | 0.14 | -0.55 | **0.04*** |
| | Trait Anxiety | 0.08 | 0.81 | 0.21 | 0.55 | 0.17 | 0.60 |
| | RSES | 0.12 | 0.64 | -0.01 | 0.96 | -0.67 | **0.02*** |
| **Meaning** | t-20 | 0.41 | 0.14 | 0.14 | 0.61 | 0.39 | 0.18 |
| | T0 | -0.72 | **0.01*** | -0.21 | 0.46 | -0.23 | 0.45 |
| | State Anxiety pre | 0.39 | 0.15 | -0.84 | **0.005*** | 0.02 | 0.92 |
| | Trait Anxiety | -0.19 | 0.55 | 0.54 | 0.11 | -0.34 | 0.33 |
| | RSES | 0.15 | 0.58 | 0.004 | 0.98 | -0.55 | **0.05*** |
| **Pragmatics** | t-20 | -0.09 | 0.71 | 0.14 | 0.65 | 0.13 | 0.66 |
| | T0 | -0.15 | 0.58 | -0.28 | 0.40 | -0.09 | 0.77 |
| | State Anxiety pre | -0.87 | **0.003*** | 0.08 | .78 | -0.54 | 0.07 |
| | Trait Anxiety | 0.68 | **0.04*** | -0.25 | 0.50 | 0.29 | 0.42 |
| | RSES | 0.11 | 0.67 | -0.17 | 0.58 | -0.36 | 0.24 |
| **Correctness** | t-20 | 0.43 | 0.16 | 0.49 | 0.11 | -0.01 | 0.95 |
| | T0 | -0.40 | 0.20 | -0.64 | **0.05*** | -0.24 | 0.41 |
| | State Anxiety pre | -0.15 | 0.61 | 0.03 | 0.90 | -0.70 | **0.01*** |
| | Trait Anxiety | -0.34 | 0.35 | 0.14 | 0.70 | 0.68 | **0.05*** |
| | RSES | -0.40 | 0.19 | -0.04 | 0.88 | -0.14 | 0.61 |
| **Translated words** | t-20 | - | - | 0.04 | 0.86 | -0.26 | 0.29 |
| | T0 | - | - | -0.38 | 0.17 | -0.15 | 0.56 |
| | State Anxiety pre | - | - | 0.06 | 0.81 | 0.18 | 0.44 |
| | Trait Anxiety | - | - | -0.27 | 0.39 | -0.61 | **0.04*** |
| | RSES | - | - | 0.14 | 0.60 | -0.80 | 0.75 |

* Significant at p < .05 level.

scored higher for pragmatic accuracy had the highest levels of cortisol immediately after finishing the translation of Text 3 (t+35) and even in the recovery phase (t+45). On the one hand, data on the detrimental effect of experimental stress on performance even in the absence of time pressure constraints align with findings that support the negative effect of stress on cognition and task performance [64–66]. On the other, the negative influence of stress on number of translated words under tight time pressure agrees with evidence pointing to the influence of time pressure on adjusting the decision process strategy based on evaluation of costs and benefits. When little time is available and the task is complex, there is a tendency to use strategies that can be rapidly applied [11]. In this sense, leaving text untranslated appears as one of the most immediate and rapid strategies. The higher post-task cortisol levels of those who performed better on pragmatic accuracy in Text 3 could be related to their levels of trait anxiety, a result that aligns with evidence suggesting that higher trait anxiety may be associated with impaired cortisol recovery in university students [67]. Considering that higher levels of trait anxiety were associated with better pragmatic accuracy under no time restrictions (Text 1), it is possible that those who performed better in Text 3 had higher levels of trait anxiety, which could have kept their cortisol levels higher during post-event processing. Besides, the fact that no traces of recuperation were found in our data during the recovery phase could be related to

the perception of uncontrollability of a task that was extremely hard to complete given the extreme time restrictions. Evidence indicates that tasks containing uncontrollable elements are associated with the longest times to cortisol recovery [41].

Our second hypothesis predicted that under time pressure constraints (i.e., Text 2 and Text 3), lower levels of self-esteem would be associated with heightened cortisol reactions, higher levels of state anxiety and impoverished performance. Conversely, higher levels of self-esteem would be associated with attenuated response of cortisol to acute time pressure, lower levels of state anxiety and enhanced performance. Results from the study confirmed our hypothesis for the expected relationship between self-esteem, anxiety and cortisol response for the preparation and recovery phases of the experimental task. Data showed that translation trainees with higher levels of self-esteem exhibited lower levels of trait anxiety, pre- and post-task state anxiety and cortisol in the preparation phase (t0) and in the recovery phase (t+45), suggesting that self-esteem is a protective factor against stress. Our data revealed that those trainees with higher pre-task levels of state anxiety also exhibited higher levels cortisol in the recovery phase (t+45), and that those with higher post-task levels of state anxiety also experienced higher cortisol levels during baseline (t-20) and recovery phase (t+45). However, no statistically significant results were reported for the relationship between self-esteem, trait and state anxiety and cortisol levels during the phases related to the translation of the texts: Text 1 (t+20) and Text 2 and Text 3 (t+35). This result somehow aligns with the previously hypothesized influence of an attentional response to the translation task and suggests that increased attention could be a protective factor against stress and emotional turmoil.

As for the influence of self-esteem on translation performance, results from the regression analysis rejected our hypothesis, pointing to higher self-esteem as a predictor of lower meaning and total accuracy in the condition with the tightest time constraints (Text 3). In addition, correlation analyses also revealed an association between higher self-esteem and higher number of translated words under moderate and tight time constraint conditions (Text 2 and Text 3). These results expand on the findings from translation studies that suggest a negative effect of self-esteem on translation performance, specifically on a higher number of spelling and punctuation errors [38]. They also align with results from psychology and health studies pointing to the effect of time pressure and task difficulty on confidence [28, 29]. On the most stringent conditions, trainees with higher self-esteem were possibly overconfident about the difficulty of the task and overestimated their abilities aiming at translating the highest number of words possible. However, given the limited time available, their effort to translate more words resulted in lesser attention to mistakes.

Our third hypothesis predicted that under time pressure constraints (i.e., Text 2 and Text 3), higher levels of self-reported anxiety will be associated with poorer performance. The regression analysis revealed that trainees with higher levels of state anxiety were more likely to make mistakes (on pragmatics in Text 1, meaning in Text 2 and correctness in Text 3), whereas those with higher levels of trait anxiety were less prone to make them (at least on pragmatics in Text 1 and correctness in Text 3). Results on trait anxiety point to a possible differential effect between trait and state anxiety and between trait anxiety and self-esteem on translation performance. Whereas higher state anxiety was found to be a predictor of less accuracy across the three conditions, trait anxiety predicted higher accuracy (at least on pragmatics and correctness) under no time pressure and also under very tight time constraints. In the latter condition (Text 3), higher self-esteem predicted less meaning and total accuracy. The result on trait anxiety aligns with findings on its beneficial effect for enhanced vigilance [68]. Increased vigilance caused by sustained anxiety could have been helpful to avoid meaning mistakes under manageable time constraints. Moreover, the beneficial effect of trait anxiety also aligns with evidence from studies on competitive sports performance when combined with other traits such

as higher confidence or perfectionism [31–33]. In contrast, pre-task state anxiety seemed to have a negative effect on the efficient processing of the same information [69].

Besides the number of errors, the number of translated words was also computed as an indicator of impoverished performance. Results from the correlations analysis showed a negative association between trait and pre-state anxiety and the number translated words in the two conditions under time constraints (Text 2 and Text 3), but in the regression analysis only higher trait anxiety was found a predictor of less translated words under extreme time constraints (i.e., Text 3). Trait anxiety displayed in fact the opposite behaviour to self-esteem, being related to less mistakes, but also to a lower number of translated words. As already argued, the beneficial effect of trait anxiety on accuracy could be related to its effects on enhanced vigilance, but also to competitive anxiety-related personality traits, such as perfectionism [70]. There is the possibility that trainees with higher trait anxiety had perfectionist tendencies and were thus more worried about achieving better accuracy than translating more words.

Results from this study provide evidence for the contradictory role of self-esteem in translation performance under time constraints. Its effects on trait and state anxiety points to self-esteem as a protective psychological resource against anxiety, but its effects on translation performance are not that clear. Under stringent time constraints, it may increase task completion even at the expense of more mistakes, but no relationship is found in the absence of time pressure. As for anxiety, the provided evidence points to the detrimental effect of state anxiety on performance but to the beneficial effects of trait anxiety. Results also suggest a possible differential effect between trait anxiety and self-esteem on translation performance: whereas higher self-esteem is found to be a predictor of less accuracy under the acute time pressure condition, trait anxiety levels emerge as a predictor of less errors under the same condition, the latter being possibly a consequence of increased vigilance and/or association with personality traits leading to better translation performance [39, 40]. Nevertheless, the present study is exploratory and further research needs to be conducted to elucidate the factors that may determine the differential effect of these two psychological states on translation performance under different time pressure conditions.

There are also some obvious limitations to our research that should be addressed in future studies. Firstly, limitations on our sample should be acknowledged, since 33 participants may not be a high enough number to detect definite trends. Even if within-subject designs have greater statistical power than between-subject ones, the study should be replicated with more participants and different populations. Results from trainees with different levels of translation competence and different working languages should be compared. Professionals should also be tested to compare the effects of different levels of expertise.

Secondly, even if the three texts used as stimuli were highly comparable and all participants translated the same text under the same time pressure conditions, there is still the possibility that text differences played a role in final quality performance. A between-subjects design would allow us to rule text differences out, but could bring in individual differences in psychological traits, translation competence and speed. Nonetheless, future counterbalanced designs with bigger samples are needed to isolate the main effects of time and text.

Thirdly, cortisol response is a reliable indicator of stress, but cortisol response is also variable and very sensitive to a number of factors, such as participants' histories of medical or psychological disorders or, in the case of women, being on the pill or even the phase of their menstrual cycle. No cases of medical or psychological disorders were found in our sample. As previously mentioned, female participants were asked for their phase of the menstrual cycle, but since most of them admitted not being sure of their answers, the variable was not included in the study. The four males recruited in the sample and the eight women who reported being

on hormonal contraceptives were finally included in our analysis, since their exclusion did not alter the pattern of cortisol response, which still showed a decreasing trend across the tasks. Phases of the menstrual cycle should, nevertheless, be controlled for in future studies to discard their potential effect. Similarly, our study did not control for sex differences because only four men were included in our sample and our design focused on within-subject differences, but future designs accounting for these differences should be carried out.

Finally, the use of additional physiological measures would be useful to better characterize the autonomic and cognitive activity supporting cortisol response. The combination of cortisol with other measures such as HRV or even eye-tracking would be advisable to determine in future studies whether the reported pattern of cortisol response was due to stress by time pressure or to the allocation of attentional resources during the translation task.

## Conclusions

Data from self-report measures in the present study shed some light on the role of self-esteem and anxiety in translation performance under time pressure conditions. Results point to self-esteem as a protective factor against stress and trait anxiety as a predictor of higher accuracy. Translation trainees with higher self-esteem exhibited lower levels of anxiety and cortisol in the preparation and recovery phases. As for translation performance, under extreme time pressure conditions, higher self-esteem was a predictor of higher number of errors. In contrast, trait anxiety predicted a lower number of errors under moderate time pressure and a lower number of translated words under extreme time pressure conditions. Levels of pre-task cortisol (t-0) and state-anxiety also emerged as predictors of a higher number of errors under no time constraints and moderate time pressure. Additionally, trainees with the highest pre-task levels of cortisol and of trait and state anxiety translated less words under moderate and extreme time pressure conditions. In contrast, those with the highest levels of self-esteem translated the greatest number of words under the same conditions.

The cortisol data showed no statistically significant differences between the time conditions and displayed a decreasing pattern that pointed to the effect of an attentional response to the task that may have increased as trainees engaged with the translation. But even if unexpected, data from the physiological biomarker provided additional information about the influence of experimental stress on trainees' performance and behavior during the translation task as well as on the potential role of attention in the stress response.

Quality of work and the ability to meet deadlines are two essential assets for any employee or freelancer, and translators are no exception. Modern technology, increasing competitiveness and immediacy of communication have fueled translation needs but also intensified time pressure for provision of translation services. Translations are needed on the fly and translators are expected to work fast and well. Translation students are trained to work swiftly and efficiently, favoring multitasking and Computer Assisted Translation (CAT) tools. Competition has increased rapidly and dramatically in the translation market. But time pressure has already started to take a toll on trainee and professional translators' physical and mental health. Translation blogs are increasingly including references to work-related mental health issues, such as anxiety, stress or burnout syndrome. Tips for managing these conditions usually include coping skills and awareness of one's personal needs and limits, but knowledge of their causes and effects would also be of help. The results of this study can contribute to this knowledge by providing useful information about the causes and effects of time pressure on translation performance and about the modulatory role of self-esteem and anxiety. Results remind translators of the relevance of understanding the mechanisms mediating quality performance in highly demanding situations, such as working under extreme time pressure. Competitive

translators must have the required expertise to guarantee maximum efficiency and productivity even in the face of adverse situations.

## Supporting information

**S1 File. Three translation tasks.**
(DOCX)

## Author Contributions

**Conceptualization:** Ana Mª Rojo López, Paula Cifuentes Férez.

**Data curation:** Paula Cifuentes Férez, Laura Espín López.

**Formal analysis:** Laura Espín López.

**Funding acquisition:** Ana Mª Rojo López.

**Investigation:** Paula Cifuentes Férez.

**Methodology:** Ana Mª Rojo López, Laura Espín López.

**Resources:** Laura Espín López.

**Supervision:** Ana Mª Rojo López.

**Visualization:** Paula Cifuentes Férez.

**Writing – original draft:** Ana Mª Rojo López, Paula Cifuentes Férez.

**Writing – review & editing:** Ana Mª Rojo López, Paula Cifuentes Férez, Laura Espín López.

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
