## [Decision Letter · Decision Letter 0]

23 Jul 2021

PONE-D-21-14186

The influence of time pressure on translation trainees’ performance: testing the relationship between self-esteem, salivary cortisol and subjective stress response

PLOS ONE

Dear Dr. Rojo,

Thank you for submitting your manuscript to PLOS ONE. After careful consideration, we feel that it has merit but does not fully meet PLOS ONE’s publication criteria as it currently stands. Therefore, we invite you to submit a revised version of the manuscript that addresses the points raised during the review process.

I have reports from a knowledgeable referee who raises questions about an important part of the experimental design. I am offering you a chance to revise the paper in light of their concern. No promises, of course.

We look forward to receiving your revised manuscript.

Kind regards,

Joydeep Bhattacharya

Academic Editor

PLOS ONE

Additional Editor Comments:

I have reports from a knowledgeable referee who raises questions about an important part of the experimental design. I am offering you a chance to revise the paper in light of their concern. No promises, of course.

Reviewers' comments:

Reviewer's Responses to Questions

**Comments to the Author**

1. Is the manuscript technically sound, and do the data support the conclusions?

Reviewer #1: Partly

2. Has the statistical analysis been performed appropriately and rigorously? 

Reviewer #1: Yes

3. Have the authors made all data underlying the findings in their manuscript fully available?

Reviewer #1: Yes

4. Is the manuscript presented in an intelligible fashion and written in standard English?

Reviewer #1: Yes

5. Review Comments to the Author

Reviewer #1: Comments on the manuscript “The influence of time pressure on translation trainees’ performance: testing the relationship between self-esteem, salivary cortisol and subjective stress response”.

The goals of this project are interesting and there is an information gap on the cortisol response in translation trainees’ performance that justifies the need to carry out this project. However, I have some doubts about possibility of publishing the manuscript as it is written and interpreted.

My main concern is with the experimental design. For example, the first hypothesis was: “the less time available to produce the translation, the highest the students’ cortisol reactivity to time pressure…” but How can the authors could measure the effect of time pressure on cortisol if they did not measure it as a separate design? i.e., one group each condition. If participants were told in the consent form that they were to translate three different texts, they could’ve been predisposed to do so. On the other hand, in this within subject design, why not take a saliva sample after the text 2 condition? By taking a saliva sample after the two time pressures, masks the effect of both texts conditions. Other strategy could’ve been randomized the conditions to evaluate the difference between starting without the stress of time pressure or starting with the stress of time pressure. For instance, on Line 363 Result’s section: To examine whether self-esteem, stress-induced cortisol response and anxiety could be associated with translation performance scores for the three text… Again, if the authors had examined only (or separated) the effect of different time pressures, it would be correct to say stress, but starting with a task without the stress of time pressure, that situation surely is not stress.

Additionally, the STAI-S questionnaire applied at the end of the procedure, measured the hole task emotional state, not only one condition.

I don’t know whether the authors could rethink the hypothesis and goals, as they are, did not explain the effect of time pressure on cortisol and anxiety.

Regarding the Introduction section, it is concentrated on time pressure and decision-making theories, but less space is dedicated to explain the implications of cortisol response, and the relationship of cortisol with self-esteem. There are some interesting articles dedicated to that relationship (e.g., Yang, J., Yang, Y., Li, H., Hou, Y., Qi, M., Guan, L., ... & Pruessner, J. C. 2014. Correlation between self-esteem and stress response in Chinese college students: The mediating role of the need for social approval. Personality and Individual Differences, 70, 212-217).

Minor comments:

Line 63-71. It would be more appropriated -and organized-, to cite first the positive effects of stress on performance and then the negative, or vice versa.

Line 347. How can you explain the similar correlation (-0.40) between self-esteem and t0 and self-esteem and t+45 with different p value (<0.002 and <0.001).

Line 424. I don’t think this citation be comparable, elite athletes have another preparation than second grade trainee’s translators.

435-438. I’m not sure that you measure a recovery, the graphic that you show indicates a decrease of cortisol levels, maybe the authors should refer it only as high or low levels

In the Method section, it would be more appropriated to start explaining the inclusion and exclusion criteria, as it is written looks disorganized. Did you approach to the participants? Or did you announce the protocol?

What was the internal consistency of the instruments applied to the sample? For instance, citation 51 corresponds to US citizens.

Line 261: You said that five saliva smaples were taken; later, on line 270 you wrote “all four samples”

Line 277. The authors wrote “the experimental session took place in a quiet room at the Faculty of Arts”, and later, on line 286, “After the arrival at the laboratory”

Line 298. It is not understood what it means: On completion the 3rd cortisol sample was taken.

6. PLOS authors have the option to publish the peer review history of their article (what does this mean?). If published, this will include your full peer review and any attached files.

Reviewer #1: **Yes: **Ana Lilia Cerda-Molina

---

## [Author Response · Author response to Decision Letter 0]

7 Aug 2021

We would like to thank the reviewer for her comments and contributions to our manuscript. We deeply appreciate her insightful comments and have incorporated the suggested changes into the manuscript to the best of our ability. We think that the quality of the manuscript has significantly improved thanks to her thoughtful and incisive revision suggestions. Specific suggestions are addressed below and justifications for our changes and answers are also provided. Please bear in mind that line numbers correspond to lines in the version of the manuscript with track changes.

Reviewer #1: 

1. The goals of this project are interesting and there is an information gap on the cortisol response in translation trainees’ performance that justifies the need to carry out this project. However, I have some doubts about possibility of publishing the manuscript as it is written and interpreted. My main concern is with the experimental design. For example, the first hypothesis was: “the less time available to produce the translation, the highest the students’ cortisol response to time pressure…” but How can the authors could measure the effect of time pressure on cortisol if they did not measure it as a separate design? i.e., one group each condition.

- We thank the reviewer for her interest in our work and her observation. We agree that the first hypothesis can be formulated with greater precision in relation to our aims. Our previous formulation seemed to place the focus on the cortisol pattern across the three different translations. However, our aim is rather to investigate the difference between the effects of time pressure on translation performance when translating without vs. with time restrictions. We have thus reformulated our first hypothesis as follows: “When translating under time pressure students’ cortisol levels will be higher and their performance will be worse than under no time pressure.”

- We also agree with the reviewer that using a separate design with one group in each condition could in a way facilitate the measurement of the effect of time pressure on cortisol. However, we purposely chose a within-participant design to minimize differences in the participants’ translation competence and styles. Even when students belong to the same academic year, differences in their levels of competence are often quite prominent, being a relevant intervening variable which can mask the effects of the independent variable.

- There is also the possibility (something which the reviewer also suggests further on) of adding a post-task cortisol measure after Text 2. However, as explained further below, this measure was not included because the time difference between the end of Text 2 and the end of Text 1 and Text 3 is not long enough to provoke a significant cortisol response (i.e., 10 minutes between the end of Text 1 and Text 2 and 5 minutes between the end of Text 2 and the end of Text 3). Cortisol is widely used as a measure of activity and response of HPA axis to stress. The cascade-like nature of the HPA axis with two hormones – CRH (corticotropin-releasing hormone) and ACTH (adrenocorticotropic hormone) – preceding cortisol release is relatively slow, as well as is the signal transported by blood, and results in cortisol increases occur only 20–30 min after stress onset. Depending on the exact length of the stress protocol, this translates into a maximum cortisol increase being observed 30 min post-stress (Kirschbaum et al., 1993). Taking all of this into account, we chose 15 min as the minimum threshold to guarantee the increase.

2. If participants were told in the consent form that they were to translate three different texts, they could’ve been predisposed to do so. 

- We also thank the reviewer for this comment. We agree that telling them about the number of texts could somehow have predisposed them to be ready to translate 3 texts. However, we do not think this information is likely to have a significant impact on our results. Our students are used to translating texts much longer that the 3 texts used as stimuli together. Besides, they were given no details about their length, topic or level of difficulty. Moreover, our aim was not related to the number of texts but rather to the difference between translating with time restrictions or without. 

3. On the other hand, in this within subject design, why not take a saliva sample after the text 2 condition? By taking a saliva sample after the two time pressures, masks the effect of both texts conditions. 

- This is a very good point and something we initially considered doing, since it is the logical research design. Nevertheless, as already explained, the time difference between the end of Text 1 and the end of Text 2 was only 10 minutes, which did not seem long enough to cause any cortisol response, especially considering that Text 1 was not supposed to cause a significant cortisol response since no time limit was given. As stated before, our decision was supported by studies which advise to have a minimum time of 10-30 minutes for cortisol responses to arise after the stressor (Kirschbaum et al., 1993; Goodman et al., 2017). The same applies to the time difference between the end of Text 2 and the end of Text 3, which was only 5 minutes. Given these limitations, we decided to take a saliva sample after translating both texts, which gave us 15 minutes for the expected cortisol response. 

- We agree with the reviewer that having only a sample after the two texts could mask differences in the cortisol response between the two stimuli under the time limit condition. Still, it allowed us to investigate our main aim, i.e., the difference between translating under no time limits versus translating under time restrictions. However, we totally understand and share the reviewer’s concern and we have inserted a brief justification of our decision in the text (315-321).

4. Other strategy could’ve been randomized the conditions to evaluate the difference between starting without the stress of time pressure or starting with the stress of time pressure. For instance, on Line 363 Result’s section: “To examine whether self-esteem, stress-induced cortisol response and anxiety could be associated with translation performance scores for the three text”… Again, if the authors had examined only (or separated) the effect of different time pressures, it would be correct to say stress, but starting with a task without the stress of time pressure, that situation surely is not stress.

- We thank the reviewer once again for her comment. This is also a very good point we initially considered doing. However, we chose not to randomize the conditions for two major reasons: to ensure the progressive build-up of time pressure (as already indicated in the text, lines 180-181); and to avoid the time limitations on cortisol response previously exposed (this information has also now been added on line 182). When having to translate Text 2 or Text 3 in the first place, the time for cortisol response (10 m. for Text 2 and 5 m. for Text. 3) would have not been long enough to provoke a significant cortisol response. Moreover, this type of randomization would make more sense when aiming to detect differences in cortisol patterns across different times or even to compare different times to determine the best condition for high quality translation work. However, our aim here was limited to test whether imposing time restrictions raises students’ levels of stress and has an effect on translation performance in comparison with having no time restrictions. 

- Regarding her comment on the lack of precision of our statement on line 363 Result’s section (now line 390) we agree that the formulation is not clear enough and we have deleted “stress-induced” from the sentence, since the cortisol levels and levels of anxiety during the translation of Text 1 are not likely to be due to stress by time pressure. In our experimental design, the translation of Text 1 is used as the control condition, since no time restrictions were given and thus no stress due to time pressure was assumed. However, this does not mean that students were not experiencing any stress. On the contrary, results showed that their stress levels were also high in that condition, most likely due to experimental or even task-related stress.

5. Additionally, the STAI-S questionnaire applied at the end of the procedure, measured the hole task emotional state, not only one condition.

- Once again, we understand the reviewer’s concern about only completing the STAI-S questionnaire before and after the procedure. We are aware that when used at the end of the procedure it measures the emotional state caused by the whole task, but we faced the same protocol limitations as with cortisol. Asking them to complete the STAI-S test after Text 1 was not relevant since no time limit was given and no stress by time pressure was assumed, and completing it between Text 2 and Text 3 would have involved only a 5 min difference with the STAI-S to be completed at the end of the procedure. Besides, two measures seemed enough considering that the STAI test and cortisol response were used, respectively, as subjective and objective validations of the stressful nature of translating under time pressure conditions.

6. I don’t know whether the authors could rethink the hypothesis and goals, as they are, did not explain the effect of time pressure on cortisol and anxiety.

- We thank the reviewer for her suggestion, which we will certainly bear in mind for future studies advancing on our present findings. We understand her concerns about the fact that our hypotheses and goals may not fully explain the effect of time pressure on cortisol and anxiety. In fact, we have reformulated Hypothesis 1 in line with her comments. Nevertheless, as previously stated, this study focuses on the effect of time pressure on translation performance. Cortisol and anxiety levels are not our final aim, but rather convenient control measures to ensure that stress increases when translating with time restrictions as compared with translating with no time limits.

7. Regarding the Introduction section, it is concentrated on time pressure and decision-making theories, but less space is dedicated to explain the implications of cortisol response, and the relationship of cortisol with self-esteem. There are some interesting articles dedicated to that relationship (e.g., Yang, J., Yang, Y., Li, H., Hou, Y., Qi, M., Guan, L., ... & Pruessner, J. C. 2014. Correlation between self-esteem and stress response in Chinese college students: The mediating role of the need for social approval. Personality and Individual Differences, 70, 212-217).

- We thank the reviewer for the reference. We have expanded our information on results on self-esteem and cortisol response and added a couple of references (including the one provided) to the literature review in the introduction (lines 146-151).

Minor comments:

8. Line 63-71. It would be more appropriated -and organized-, to cite first the positive effects of stress on performance and then the negative, or viceversa.

- We value and understand the reviewer’s recommendation, but given the scarce findings available on translation, we have opted for using the field or domain as the main criterion to present previous results, i.e., first the effects of stress on psychological health (the negative (line 60-61), the positive (line 62-63) and the mixed effects, depending on the stress intensity (68-71)). Secondly, we focus on the effects of stress on task performance (72-84) and then on the translation process. 

9. Line 388. How can you explain the similar correlation (-0.40) between self-esteem and t0 and self-esteem and t+45 with different p value (<0.002 and <0.001).

- We are grateful to the reviewer for pointing this out. We have checked our results again and verified that it was an error. Figures have been corrected in the text and in the corresponding table (Table 4).

10. Line 424. I don’t think this citation be comparable, elite athletes have another preparation than second grade trainee’s translators.

- We thank the reviewer for her comment. We totally agree that elite athletes and translation trainees have different preparation. In fact, our intention was not to state they are similar. Rather, our comparison was supported by Ericsson’s (2000) comparison between interpreting and other high performance domains, such as sports, medicine, chess or music. Ericsson’s expert-performance approach claims that in all the aforementioned domains expert performance is shown to be primarily acquired through the engagement in designed training activities, namely deliberate practice (Ericsson et al., 1993). Nevertheless, we understand that sports and translation tasks may not be comparable in terms of cortisol response and have deleted all the references to the study on cortisol response on elite athletes (reference 59 on lines 452-454, and reference 63 on line 480). We have also deleted the comparison from the end of the conclusions (references 68 and 69 on lines 641-651)

11. 435-438. I’m not sure that you measure a recovery, the graphic that you show indicates a decrease of cortisol levels, maybe the authors should refer it only as high or low levels.

- We totally agree with the reviewer’s observation. In fact, our data show no traces of recovery. We have consequently reformulated the statements referring to a recovery and referred to it only as high or low levels. We have also indicated this lack of recovery in the text (lines 484-485). 

12. In the Method section, it would be more appropriated to start explaining the inclusion and exclusion criteria, as it is written looks disorganized. Did you approach to the participants? Or did you announce the protocol?

- We thank the reviewer for pointing this out to us. We have reformulated the beginning (lines 186-192) to clarify the procedure and the inclusion and exclusion criteria. 

13. What was the internal consistency of the instruments applied to the sample? For instance, citation 51 corresponds to US citizens.

- The internal consistency of the Rosenberg test was 0.76 (line 268) and the STAI-T 0.92 and STAI-S 0.94 (line 274). This information has been included in the text. 

14. Line 261: You said that five saliva samples were taken; later, on line 270 you wrote “all four samples” 

- Thank you very much for pointing the mistake out. We have corrected the wrong statement and specified “five samples” as the correct number of samples taken (line 286).

15. Line 277. The authors wrote “the experimental session took place in a quiet room at the Faculty of Arts”, and later, on line 286, “After the arrival at the laboratory”

- Thank you very much for pointing the mistake out. We have replaced laboratory with a university room (line 302).

16. Line 298. It is not understood what it means: On completion the 3rd cortisol sample was taken.

- We thank the reviewer for indicating this to us. We have rephrased the sentence to “On completion of this translation the 3rd cortisol sample was taken” (line 314)

Other actions

Besides the outlined changes, we have also uploaded the two Figures onto the Preflight Analysis and Conversion Engine (PACE) digital diagnostic tool so they meet your requirements.

As far as uploading our data onto a public repository, we have uploaded them onto the Webpage of our research project EMOTRA under the Data section, which is hosted on the University of Murcia website (https://www.um.es/emotra/data/). 

We sincerely thank the reviewer once again for her work and hope that this final version will meet her requirements. The reviewer has brought up many good points and we appreciate the opportunity to improve our paper and clarify our research results.

---

## [Editor Report · Decision Letter 1]

9 Sep 2021

The influence of time pressure on translation trainees’ performance: testing the relationship between self-esteem, salivary cortisol and subjective stress response

PONE-D-21-14186R1

Dear Dr. Rojo,

We’re pleased to inform you that your manuscript has been judged scientifically suitable for publication and will be formally accepted for publication once it meets all outstanding technical requirements.

Kind regards,

Joydeep Bhattacharya

Academic Editor

PLOS ONE
---

## [Editor Report · Acceptance letter]

22 Sep 2021

PONE-D-21-14186R1 

The influence of time pressure on translation trainees’ performance: testing the relationship between self-esteem, salivary cortisol and subjective stress response 

Dear Dr. Rojo López:

I'm pleased to inform you that your manuscript has been deemed suitable for publication in PLOS ONE. Congratulations! Your manuscript is now with our production department. 

Kind regards, 

on behalf of

Dr. Joydeep Bhattacharya 

Academic Editor

PLOS ONE